# Expression of Preferentially Expressed Antigen in Melanoma, a Cancer/Testis Antigen, in Carcinoma In Situ of the Urinary Tract

**DOI:** 10.3390/diagnostics13243636

**Published:** 2023-12-10

**Authors:** Shota Fujii, Mitsuaki Ishida, Kazumasa Komura, Kazuki Nishimura, Takuya Tsujino, Tomohito Saito, Yohei Taniguchi, Tomohiro Murakawa, Haruhito Azuma, Yoshinobu Hirose

**Affiliations:** 1Department of Pathology, Osaka Medical and Pharmaceutical University, 2-7, Daigaku-machi, Takatsuki City 569-8686, Osaka, Japan; ompu20201092@s.ompu.ac.jp (S.F.);; 2Department of Urology, Osaka Medical and Pharmaceutical University, 2-7, Daigaku-machi, Takatsuki City 569-8686, Osaka, Japan; 3Translational Research Program, Osaka Medical and Pharmaceutical University, 2-7, Daigaku-machi, Takatsuki City 569-8686, Osaka, Japan; 4Department of Thoracic Surgery, Kansai Medical University, 2-5-1, Shinmachi, Hirakata 573-1010, Osaka, Japan

**Keywords:** urothelial carcinoma, carcinoma in situ, preferentially expressed antigen in melanoma

## Abstract

Carcinoma in situ (CIS) of the urinary tract comprises 1–3% of all urothelial malignancies and is often a precursor to muscle-invasive urothelial carcinoma (UC). This study aimed to examine the expression profiles of preferentially expressed antigen in melanoma (PRAME), a cancer/testis antigen, and assess its diagnostic and therapeutic applications in CIS, given that its expression in UC has been minimally studied and has not yet been analyzed in CIS. We selected consecutive patients with CIS who underwent biopsy and/or transurethral tumor resection at the Osaka Medical and Pharmaceutical University Hospital. Immunohistochemical staining for PRAME and p53 was performed. Overall, 53 patients with CIS (6 females and 47 males) were included. Notably, PRAME expression was observed in 23 of the 53 patients (43.4%), whereas it was absent in the non-neoplastic urothelial epithelium. Furthermore, no correlation was found between PRAME expression and aberrant p53 expression. Therefore, PRAME expression may serve as a useful marker for CIS of the urinary tract. Furthermore, PRAME may be a candidate for the novel therapeutic target for standard treatment-refractory CIS patients.

## 1. Introduction

Urothelial carcinoma (UC) is one of the common causes of cancer-related death worldwide. Most UCs show papillary proliferation, while carcinoma in situ (CIS) is defined as a flat neoplastic proliferation of high-grade neoplastic urothelial cells without papillary formation, comprising approximately 1–3% of all urothelial malignancies [1]. Notably, CIS is considered to carry a high risk for the development of muscle-invasive UC, as over 80% of untreated patients with CIS progress to muscle-invasive UC [2]. Therefore, early detection and treatment of CIS are among the important issues in the field of urological oncology.

As CIS does not form papillary tumors, the standard method for diagnosing CIS of the urinary tract is biopsy from all suspicious erythematous areas and/or random biopsies from the normal-appearing mucosa by cystoscopic examinations [2]. Furthermore, urine cytology has been recognized as the standard and valuable diagnostic tool for detecting high-grade UC and CIS [2] because it represents the highest predictive value in diagnosing high-grade UC. Importantly, one report demonstrated that the urine cytological examination showed a sensitivity and specificity of 87.1% and 63%, respectively, in patients with CIS [3]. Another study showed a sensitivity of 54.5% using urine cytology for detecting CIS [4]. Therefore, a significant number of patients with CIS yield negative results in cytological examinations. Accordingly, several ancillary methods, including fluorescence in situ hybridization, have been developed either alone or in combination with urine cytological examination to improve the detection of carcinoma cells. Moreover, recent studies demonstrated the usefulness of a DNA methylation test for detecting high-grade UC [5,6,7]. Reportedly, the bladder EpiCheck test^®^ analyzed the methylation status of 15 selected genes using urine specimens, and showed a high sensitivity for detecting high-grade UC [5,6,7]. Although some of these markers aid in improving the sensitivity of detection of carcinoma cells, they have not yet been universally used because of the cost and false-negative results [2,5]. Therefore, the development of a novel and convenient diagnostic marker is required to detect UC cells, especially CIS.

Preferentially expressed antigen in melanoma (PRAME) is one of the cancer/testis antigens. These antigens are a family of tumor-associated antigens expressed in malignant tumors but not in normal adult tissues, except for the testis. PRAME was first reported as a tumor-associated antigen expressed in malignant melanoma recognized by cytotoxic T lymphocytes [8]. The PRAME gene is located on chromosome 22 (22q11.22) and contains leucine-rich repeat domains [9]. This protein acts as a repressor of the retinoic acid receptor pathway, leading to the regulation of cell growth, differentiation, and apoptosis [10,11,12]. Importantly, its expression is fundamentally restricted in the non-neoplastic testis, specifically spermatogonia, and proliferative endometrial glands; it is regulated at very low levels by DNA methylation in most of the non-neoplastic tissues excluding the testis and endometrium [8,13]. Recently, PRAME expression has received much attention in some types of malignant tumor as a useful diagnostic marker due to its nature of being a cancer/testis antigen [13]. Notably, malignant melanoma and its mimics represents the most intensively investigated area of PRAME expression [14,15,16,17,18,19,20,21,22,23,24]. Numerous articles have examined the expression profiles of PRAME in malignant melanoma and have demonstrated its diagnostic utility. Specifically, most cases of malignant melanoma exhibited positive immunoreactivity to PRAME; in contrast, most benign melanocytic nevi did not [14,15,16,17,18,19,20,21,22,23,24]. Moreover, PRAME expression has also been reported in various types of malignant tumor, other than malignant melanoma, including myxoid liposarcoma, synovial sarcoma, neuroblastoma, renal cell carcinoma, non-small-cell lung cancer, and thymic cancer [8,13,14,25,26,27,28,29,30]. The expression of PRAME in UC has been analyzed in only three studies, wherein it was reported in 7%, 15%, and 20% of patients with UC, as determined by immunohistochemistry or RT-PCR [13,31,32]. However, the expression profiles of PRAME in CIS of the urinary tract have not yet been examined.

Furthermore, due to the nature of the cancer/testis antigens of PRAME, this protein has been recognized as a potential target for immunotherapy in patients with some types of malignant tumor, such as uveal malignant melanoma and non-small-cell lung cancer [33,34]. Gezgin et al. showed that PRAME-specific T cells reacted against PRAME-positive uveal malignant melanoma cell lines, and approximately 45% of primary uveal malignant melanoma showed positive immunoreactivity for PRAME [33]. Thus, PRAME may be a potential candidate for direct immunotherapy for patients with uveal malignant melanoma expressing PRAME [33]. In another study, a phase I dose escalation study for patients with PRAME-positive resected non-small-cell lung cancer was performed [34]. Recombinant PRAME proteins were injected for measuring the anti-PRAME humoral response in the study, and PRAME-specific CD4-positive T cells were detected after immunizations in all patients, but not CD8-positive T cells [34]. Only low-grade adverse events, such as injection site reactions and fever, were observed; therefore, PRAME might be a promising candidate for immunotherapy for patients with non-small-cell lung cancer [34]. These results indicate that PRAME may be a potential immunotherapeutic target for all patients with PRAME-expressing malignant tumors, independent of histological types. However, the PRAME expression profiles of UC have not yet been analyzed in detail, and it has not been elucidated whether or not PRAME is also a potential target for immunotherapy for patients with UC, especially CIS.

Accordingly, the present study aimed to analyze the immunohistochemical expression profiles of PRAME in CIS of the urinary tract and discuss the potential diagnostic and therapeutic applications.

## 2. Materials and Methods

### 2.1. Patient Selection

We selected consecutive patients with CIS who underwent biopsy and/or transurethral/surgical resection at the Osaka Medical and Pharmaceutical University Hospital between January 2021 and December 2022. Accordingly, 53 patients were included in this study.

This retrospective, single-institution study was conducted according to the principles of the Declaration of Helsinki, and the study protocol was approved by the Institutional Review Board of Osaka Medical and Pharmaceutical University (Approval #2020-124 and #2023-027). All data were anonymized. The institutional review board waived the requirement for informed consent because of the retrospective study design, as medical records and archived samples were used with no risk to the participants. Moreover, the present study did not include minors. Information regarding this study, such as the inclusion criteria and the opportunity to opt out, was provided through the institutional website (https://www.ompu.ac.jp/u-deps/path/img/file9.pdf) (accessed on 7 December 2023).

### 2.2. Histopathological Analysis

Biopsied and/or resected specimens were fixed with 10% buffered formalin, sectioned, and stained with hematoxylin and eosin. A review of the histopathological features was performed by two researchers (SF and MI). The diagnostic criteria for CIS were based on the descriptions provided in the WHO Classification of Tumours [1]. Cases showing the typical histopathological features of CIS were selected, and questionable cases of patients with reactive atypia or dysplasia were excluded from this study. Moreover, cases of invasive UC featuring a CIS component were excluded from this study.

### 2.3. Immunohistochemical Analyses

Immunohistochemical staining was performed using autostainers (Discovery Ultra System; Roche Diagnostics, Basel, Switzerland; Leica Bond-III; Leica Biosystems GmbH, Nußloch, Germany) according to the manufacturer’s instructions. A mouse monoclonal antibody against p53 (DO-7; DAKO; Agilent Technologies, Inc., Santa Clara, CA, USA, diluted 1:50) and a rabbit monoclonal antibody against PRAME (EPR20330; Abcam, Cambridge, UK, diluted 1:100) were used. Whole biopsied or resected specimens of CIS were used for immunostaining, and no tissue microarray was applied in the present study. A minimum of two researchers independently evaluated the immunohistochemical staining results.

Positive immunoreactivity for PRAME, manifesting as either nuclear and/or cytoplasmic positivity, was defined as any expression in carcinoma cells of CIS, employing the same methodology as that outlined in the previous report [31]. Similarly, p53 immunoreactivity was assessed in accordance with the previous report [35], as follows: aberrant patterns included either diffuse positive (with the majority of carcinoma cells exhibiting homogeneous moderate to strong positivity) or negative/null patterns, while wild-type patterns featured patchy positive cells.

### 2.4. Statistical Analysis

The correlation between the two groups was analyzed using Fisher’s exact test. A *p*-value < 0.05 was considered statistically significant.

## 3. Results

### 3.1. Patient Characteristics

This study included 53 patients (6 female and 47 male). The median age was 73 years (range: 47–92 years). The lesion locations were the urinary bladder and ureter in 50 and 3 patients, respectively. 

### 3.2. Histopathological and Immunohistochemical Features

Typical histopathological features of CIS are shown in Figure 1. Flat proliferation of the neoplastic urothelial cells with large round to oval nuclei with or without conspicuous nucleoli was noted. The nuclei size was more than three or four times larger than that of the non-neoplastic urothelial cells, and these nuclei showed an irregular contour and hyperchromasia. Mitotic figures were observed in most cases. No invasive neoplastic growth was observed in any of the lesions.

Figure 2 shows a typical immunohistochemical expression for PRAME in CIS. PRAME expression was noted in 23 of 53 patients (43.4%), but the non-neoplastic urothelial epithelia around the CIS lesions showed no positive immunoreactivity for PRAME in all cases (*p* < 0.001). Positive immunoreactivity for PRAME in CIS was noted in the nuclei and/or cytoplasm of the neoplastic cells of CIS, mostly only in the nuclei of the neoplastic cells. Aberrant and wild-type patterns of p53 expression are shown in Figure 3.

Aberrant p53 expression (diffuse positive or null patterns) was noted in 29 patients (54.7%) and a wild-type pattern was noted in 24 patients (45.3%). 

No correlation was noted between PRAME expression and p53 expression patterns (Table 1, *p* = 0.82).

## 4. Discussion

The expression of PRAME has been reported in various types of malignant tumor [8,13,14,15,16,17,18,19,20,21,22,23,24,25,26,27,28,29,30]. Notably, the present study is the first study to clearly demonstrate that PRAME was expressed in 43.4% of CIS of the urinary tract. Only three previous studies addressed the expression profiles of PRAME in UC. Table 2 summarizes the expression profiles of PRAME expression in UC of the previous studies, as well as that of the present study. An immunohistochemical study using tissue microarrays revealed that only 7% of UC (the information regarding histological grade and whether it was invasive or non-invasive were not available) showed positive immunoreactivity for PRAME [13]. In another immunohistochemical study using tissue microarrays, PRAME expression was noted in 15% of muscle-invasive high-grade UC [31]. Furthermore, an RT-PCR assay showed PRAME expression in 20% of non-invasive papillary and invasive UC cases [32]. Importantly, PRAME expression was observed in only 3% of non-invasive papillary UC, compared to 16% in pT1 invasive UC and 28% in pT2-4 invasive UC [32]. Moreover, although a significant correlation was found between the presence of CIS and PRAME expression in patients with non-muscle invasive UC, information regarding PRAME expression, specifically regarding the CIS component, was not available [32]. According to the results of these previous studies, PRAME expression is not a rare finding, especially in muscle-invasive and higher histological grade UC, although the detection methods and primary antibody used in immunohistochemical analyses were not the same [13,31,32]. In the present study, we demonstrated that 43.4% of CIS of the urinary tract showed positive immunoreactivity for PRAME, which was a rate higher than that reported for muscle-invasive UC and non-invasive papillary UC [31,32]. This result suggests that the expression of PRAME may differ between papillary UC and CIS, and might be related to carcinogenesis of CIS of the urinary tract. Moreover, this difference might be related to the relationship between PRAME expression and presence of CIS in patients with non-muscle invasive UC [32]. 

PRAME expression is closely regulated at very low levels by DNA methylation in adult non-neoplastic tissues except the testis and endometrium [8,13]. Due to its nature as a cancer/testis antigen, PRAME expression has been recognized as a useful diagnostic marker for several types of malignant tumor, especially malignant melanomas [14,15,16,17,18,19,20,21,22,23,24]. Table 3 summarizes the typical examples of PRAME expression in the various types of non-epithelial and epithelial malignant tumors. A high frequency of PRAME expression is reported in malignant melanomas, myxoid liposarcomas, synovial sarcomas, neuroblastomas, and thymic squamous cell carcinomas [13,25,26,27,28,30,36]. PRAME expression is more frequently noted in non-epithelial malignant tumors than in epithelial malignant tumors (Table 3); moreover, its expression has been noted in various types of non-epithelial and epithelial malignant tumors at low-to-intermediate levels (Table 3) [13,25,36,37,38,39,40,41,42,43,44,45,46,47,48]. Importantly, no PRAME expression was reported in some types of malignant tumor, such as prostate cancer, gastric cancer, and gastrointestinal stromal tumors [13]. Therefore, PRAME expression might depend on the origin of the malignant tumors. PRAME expression can be readily detected using the immunohistochemical method, as demonstrated in this study, and it may be particularly useful for differentiating malignant tumors, notably malignant melanomas, from benign melanocytic nevi. However, the positive ratio of PRAME varies according to the tumor’s origin. Therefore, the origin of the tumor must be considered when PRAME expression is used in the diagnosis of malignant tumors. 

The mechanism of PRAME expression in malignant tumors has not yet been elucidated in detail. PRAME expression is reported to be regulated by some genes, such as SOX17 (SRY-box 17) and MZF1 (myeloid zing finger 1) [12]; moreover, it shows its functions via regulation of some target genes, including p53, p21, retinoic acid receptor, and S100A4, leading to cell proliferation control, differentiation, growth arrest, and apoptosis in several human malignancies [12]. For example, PRAME expression is promoted by MZF-1, which leads to the enhancement of colony-forming capability of melanoma cells, and is inhibited by miR-211 in malignant melanoma cells [12,49]. PRAME is downstream of SOX17 and LIN28 for regulating pluripotency and controlling germ cell differentiation in seminoma cells [50]. In acute myeloid leukemia cells, the overexpression of PRAME correlates with the decreased expression of Hsp27 (heat shock protein 27), S100A4, and p21 at the transcription levels [51]. Moreover, PRAME expression is associated with the cell cycle from the G0/G1 phase to the S phase, and its overexpression correlates with the suppression of apoptosis in leukemic cells [52]. Accordingly, PRAME expression may control cell proliferation, differentiation, and apoptosis via the several above-mentioned molecular pathways in some types of malignant tumor. However, detailed molecular mechanisms regarding controlling PRAME expression and its downstream pathways remains unclear; moreover, these mechanisms might be different depending the origin of malignant tumors. Recently, it has been reported that PRAME forms a complex with p14/alternate reading frame (ARF) (CDKN2A), a well-established tumor suppressor, as well as with Cullin 2 RING E3 ligases [53]. Specifically, PRAME serves as a specific receptor protein for targeting p14/ARF for degradation, and this complex consequently inhibits cell growth [53]. Notably, the downregulation of PRAME significantly impedes cancer cell growth by inducing G2/M cell cycle arrest in hepatocellular carcinoma cell lines and lung cancer cell lines [50]. This molecular mechanism may elucidate the relationship between PRAME expression and the malignant nature of tumors. However, the detailed molecular mechanisms governing the upstream signaling pathways of PRAME expression in UC, especially CIS, remain to be elucidated. Further studies are required to clarify this mechanism.

Further, the present study demonstrated that PRAME expression in CIS of the urinary tract did not correlate with p53 aberrant expression. Diffuse positive or null immunoreactivity for p53 has been considered to be one of the characteristic features of CIS of the urinary tract and, thus, has been used as a useful diagnostic marker in the setting of pathological diagnosis [35,54]. However, recent studies have demonstrated that a relatively large number of CIS cases showed wild-type expression patterns of p53 (approximately 30–60% of CIS in the previous reports and 45.3% of the present cohort) [35,54]. In the present cohort, 10 of 24 (41.7%) p53 wild-type CIS patients showed positive immunoreactivity for PRAME, and expression patterns of p53 were not used for diagnosis of CIS in these patients. Therefore, immunohistochemical staining for PRAME, especially a combination of PRAME and p53, might help to diagnose CIS of the urinary tract.

The current therapeutic options for patients with CIS are limited, and intravesical Bacillus Calmette–Guérin (BCG) treatment has been the mainstay for this subset of patients [2]. However, a modest rate of complete response to BCG has been identified as the unmet need for innovative therapeutic approaches, as such patients would be candidates for radical cystectomy [2]. Although the efficacy of novel immunotherapeutic or viral agents, including anti-PD-1 inhibitors and nadofaragene firadenovec, has been investigated for these patients [55,56,57], developing new therapeutic targets has been an urgent issue. Recently, PRAME has garnered considerable attention for its therapeutic potential in the field of oncology [11]. This protein is recognized as one of the potential targets for immunotherapy because it is a cancer/testis antigen, and the peptides on MHC class I derived from PRAME are recognized by cytotoxic T cells [8,12,33,34,58]. Additionally, some clinical studies have been performed to evaluate the usefulness of PRAME as a target for antigen-specific immunization by adoptive T-cell immunotherapy in patients with metastatic uveal malignant melanoma and non-small-cell lung cancer [33,34,58]. Moreover, PRAME expression has been reported to be correlated with chemotherapy resistance in patients with UC and some malignant lymphomas [12,32,38,59]. Thus, PRAME may serve as a candidate for immunotherapy in patients with high-grade UC, as well as in those with CIS.

There are several limitations to the present study. First, this was a single-center retrospective analysis with a relatively small number of CIS patients, potentially leading to bias in the statistical power. Therefore, additional multi-center studies with larger cohorts are needed. Second, the expression of PRAME was analyzed by the immunohistochemical method, employing the same method and antibody as carried out in a previous report [31]. RT-PCR was used for the detection of PRAME expression in UC in another study [32]. Moreover, the correlation between protein expression detected by the immunohistochemical method and mRNA expression by RT-PCR was not available in the present study. However, a significant positive correlation between immunohistochemical analysis and PCR measurement of PRAME expression in malignant lymphoma has been shown [38]. Thus, the immunohistochemical analysis may be useful and convenient for detecting PRAME expression.

## 5. Conclusions

The present study demonstrated that 43.4% of CIS of the urinary tract showed positive immunoreactivity for PRAME. Detection of PRAME may be a useful marker for CIS of the urinary tract due to the nature of the cancer/testis antigen and might be a candidate for immunotherapy for CIS patients. Further studies are needed to clarify the expression profiles of this protein in the larger cohort. Moreover, the application of PRAME expression by the less invasive diagnostic methods for CIS compared to biopsy must be studied.

## Figures and Tables

**Figure 1 diagnostics-13-03636-f001:**
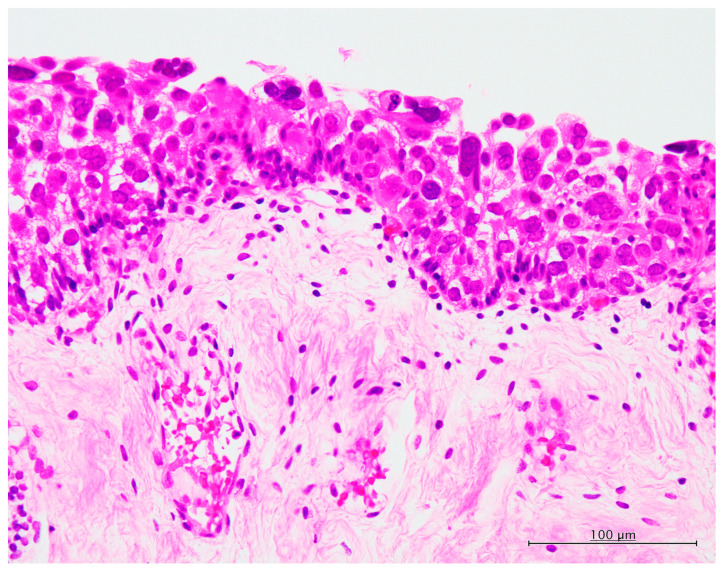
Typical histopathological features of carcinoma in situ of the urinary tract. Non-papillary flat neoplastic growth of the neoplastic cells having large round to oval nuclei (hematoxylin and eosin, ×400).

**Figure 2 diagnostics-13-03636-f002:**
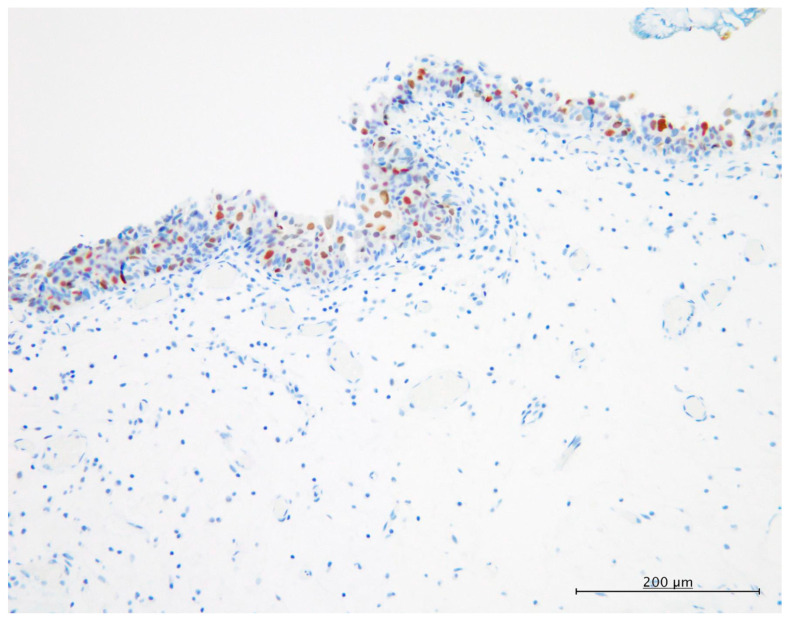
Immunohistochemical staining for PRAME. PRAME is expressed in the nuclei of the neoplastic cells of carcinoma in situ of the urinary tract (×200).

**Figure 3 diagnostics-13-03636-f003:**
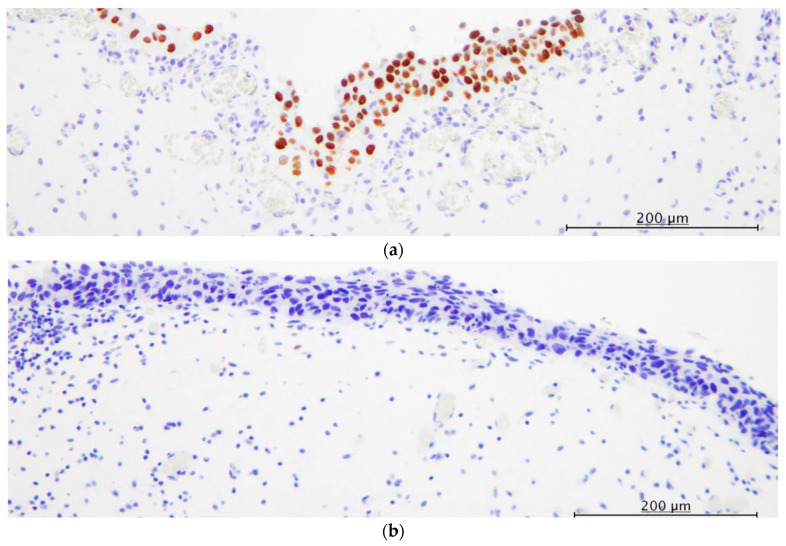
Immunohistochemical stainings for p53. (**a**) Diffuse expression pattern. (**b**) Null expression pattern. (**c**) Wild-type expression pattern (×200, each).

**Table 1 diagnostics-13-03636-t001:** Correlation between PRAME expression and p53 expression patterns in carcinoma in situ of the urinary tract.

	p53 Aberrant Pattern	p53 Wild Pattern
PRAME positive	13	10
PRAME negative	16	14

PRAME, preferentially expressed antigen in melanoma.

**Table 2 diagnostics-13-03636-t002:** Summary of PRAME expression in urothelial carcinoma.

PRAME Expression	Detection Methods	References
7% (histological grade not available)	IHC	[13]
15% of muscle-invasive high-grade UC	IHC	[31]
3% of non-invasive papillary UC 16% of pT1 UC 28% of pT2-4 UC	RT-PCR	[32]
43.4% of CIS	IHC	Present study

CIS, carcinoma in situ; IHC, immunohistochemistry; PRAME, preferentially expressed antigen in melanoma; UC, urothelial carcinoma.

**Table 3 diagnostics-13-03636-t003:** Summary of PRAME expression in various types of malignant tumor.

**Histological Type of Tumor**	**Frequency of PRAME Expression**	**References**
**Non-epithelial tumors**		
Malignant melanoma		
Skin	High	[13,14,15,16,17,18,20,21,22,23,24]
Mucosal	High	[19,25,33]
Myxoid liposarcoma	High	[13,25,26,36]
Synovial sarcoma	High	[13,25,27,36]
Neuroblastoma	High	[13,25,28]
Seminoma	Intermediate–high	[13,25,37]
Rhabdomyosarcoma	Intermediate	[13,25,36]
Malignant peripheral nerve sheath tumor	Low–intermediate	[13,25,36]
Angiosarcoma	Low–intermediate	[13,25,36]
Malignant lymphoma	Low–intermediate	[13,38]
Ewing sarcoma	Low	[13,25,36]
Leiomyosarcoma	Low	[13,25,36]
Glioblastoma	Low	[13,39]
Osteosarcoma	Low	[13]
**Epithelial tumors**		
Thymic cancer	High	[13,30]
Lung cancer	Intermediate–high	[13,25,40,41]
Breast cancer	Intermediate–high	[13,25,42,43,44]
Ovarian cancer	Intermediate	[13,25,45,46]
Renal cell carcinoma	Low–intermediate	[13,25,29]
Colorectal cancer	Low–intermediate	[13,47]
Hepatocellular carcinoma	Low	[13,48]
Pancreatic cancer	Low	[13]

PRAME, preferentially expressed antigen in melanoma.

## Data Availability

All data generated and analyzed in this study are included in this published article.

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
