# Peer review of "Expression of Preferentially Expressed Antigen in Melanoma, a Cancer/Testis Antigen, in Carcinoma In Situ of the Urinary Tract"

_diagnostics, 2023, doi:10.3390/diagnostics13243636_

Round 1
Reviewer 1 Report
Comments and Suggestions for Authors
This is an interesting study that aims to examine the expression profiles of Preferentially expressed antigen in melanoma (PRAME), a cancer testis antigen, and assess its diagnostic and therapeutic applications in carcinoma in situ (CIS) of the urinary tract. The authors analyzed immunohistochemical PRAME expression in 53 urinary CIS. The authors found a higher PRAME expression in neoplastic urothelium, whereas it was absent in the non-neoplastic one. However, how is it possible to make a diagnosis of CIS with a wild-type p53? Did the authors use other immunohistochemical stainings (CK20, CD44, ki67)? Only morphology is not sufficient to make such a diagnosis in the majority of cases. The authors should clearly explain how they made the diagnoses in such doubtful cases. Moreover, I would like the authors to use better histological images to show CIS since in those they have used it does not seem a CIS. Nonetheless, the authors should perform a statistical analysis and provide a statistically significant p value to support the association between PRAME and CIS (they only state that PRAME expression is noted in 43.4% of cases, but what is the statistical validity of such a percentage?). Notably, the sentence in the Abstract section (lines 27-28) "our findings revealed a higher positive rate for PRAME in CIS than in invasive high-grade UC" is not well explained, since such a comparison was made with literature data and not with authors' data (in fact the authors' manuscript only considers CIS and does not take into account invasive urothelial carcinomas). The authors should correct it. Of note, I think that in the introduction some literature data regarding urine methylation analyses to detect urothelial carcinoma should be presented: in fact, they represent important ancillary methods to support the urothelial carcinoma diagnosis (for example see the following references: doi: 10.1016/j.urolonc.2020.06.018; doi: 10.1016/j.humpath.2021.09.007; doi: 10.3390/jcm11133855).
Comments on the Quality of English LanguageOnly minor English editing is needed.
Author Response
December 2, 2023
Prof. Dr. Andreas Kjaer
Editor-in-Chief
Diagnostics
Resubmission - Manuscript ID: diagnostics-2656788
Dear Editor,
I would like to resubmit our manuscript titled ‘Expression of preferentially expressed antigen in melanoma, a cancer-testis antigen, in carcinoma in situ of the urinary tract’ for publication in Diagnostics.
The reviewers’ comments were highly insightful and enabled us to greatly improve the quality of our manuscript. Please find below the point-by-point responses to all the comments provided by the Reviewers. The revisions in the manuscript can be seen in red color font.
I hope that with these revisions you will find our manuscript closer to publication in Diagnostics. I look forward to hearing from you at your earliest convenience.
Sincerely,
Mitsuaki Ishida, MD, PhD.
Department of Pathology, Osaka Medical and Pharmaceutical University
2-7, Daigaku-machi, Takatsuki City, Osaka, 569-8686, Japan.
Tel: +81-72-683-1221
E-mail: mitsuaki.ishida@ompu.ac.jp
Response to the comments of Reviewer #1
Thank you very much for reviewing our manuscript. We appreciate your constructive comments. We have made the following revisions in response to the issues you raised.
- how is it possible to make a diagnosis of CIS with a wild-type p53? Did the authors use other immunohistochemical stainings (CK20, CD44, ki67)? Only morphology is not sufficient to make such a diagnosis in the majority of cases. The authors should clearly explain how they made the diagnoses in such doubtful cases.
Response: Thank you for this pertinent question. We selected cases showing typical histopathological features of CIS according to the histopathological criteria of the WHO Classification of Tumours. Cases with questionable or doubtful histology such as of reactive atypia or CIS of dysplasia or CIS were excluded from this study. The immunohistochemical stainings for CK20, CD44, and Ki-67 were not performed in this study because none of these markers is entirely specific especially in questionable lesions. We explained this in the 2.2 Histopathological analysis section of the revised manuscript.
“Cases showing typical histopathological features of CIS were selected, and questionable cases of patients with reactive atypia or CIS or with dysplasia or CIS were excluded from this study.” (line 125-128)
- I would like the authors to use better histological images to show CIS since in those they have used it does not seem a CIS.
Response: Thank you for your suggestion. We replaced the Figure 1 with a higher magnification figure showing typical histopathological features of CIS.
- the authors should perform a statistical analysis and provide a statistically significant p value to support the association between PRAME and CIS (they only state that PRAME expression is noted in 43.4% of cases, but what is the statistical validity of such a percentage?).
Response: Thank you for your comment. PRAME expression was noted in 23 of 53 patients (43.4%), and aberrant p53 expression in 29 of 53 patients (45.3%) in this cohort of CIS. The correlation between PRAME expression and p53 expression patterns in CIS are summarized in Table 1. Statistical analysis revealed no correlation between PRAME expression and p53 expression patterns in CIS.
- Notably, the sentence in the Abstract section (lines 27-28) "our findings revealed a higher positive rate for PRAME in CIS than in invasive high-grade UC" is not well explained, since such a comparison was made with literature data and not with authors' data (in fact the authors' manuscript only considers CIS and does not take into account invasive urothelial carcinomas). The authors should correct it.
Response: Thank you for your comment. We did not perform the immunohistochemical analysis of PRAME expression in invasive UC in this study. As you suggested, we omitted the description in the Abstract section.
From “Our findings revealed a higher positive rate for PRAME in CIS than in invasive high-grade UC. Therefore, its expression may serve as a useful marker for CIS of the urinary tract.”
To “Therefore, PRAME expression may serve as a useful marker for CIS of the urinary tract.” (line 27-28)
- Of note, I think that in the introduction some literature data regarding urine methylation analyses to detect urothelial carcinoma should be presented: in fact, they represent important ancillary methods to support the urothelial carcinoma diagnosis (for example see the following references: doi: 10.1016/j.urolonc.2020.06.018; doi: 10.1016/j.humpath.2021.09.007; doi: 10.3390/jcm11133855).
Response: Thank you for your suggestion. We added the comments regarding ancillary method (DNA methylation test) for diagnosis of UC in the Introduction section, and cited three references that you suggested (references 5-7).
“Moreover, recent studies demonstrated the usefulness of DNA methylation test for detecting high-grade UC [5-7]. Reportedly, bladder EpiCheck test ® analyzed the methylation status of selected 15 genes using urine specimens, and showed a high sensitivity for detecting high-grade UC [5-7]. (line 52-56)
Reviewer 2 Report
Comments and Suggestions for Authors
In this study, the authors qualitatively analyzed the expression patterns of p53 and PRAME using a cohort of 53 patients with carcinoma in situ (CIS) of the urinary tract. Specifically, this represented a pioneer study in evaluating the role of SPAME as a biomarker in the context of CIS and the complementary effects between p53 and SPAME in the diagnosis of CIS in the urinary tract. However, I do have the following concerns:
1. This is rather a small cohort, the authors should at least highlight this limitation in the discussion.
2. Figure 1 is not intuitive, I suggest authors to manually highlight those cells with distinct morphology so that readers with limited pathology background may understand.
3. What is the criterion for determining whether a patient is positive or negative using IHC?
Author Response
December 2, 2023
Prof. Dr. Andreas Kjaer
Editor-in-Chief
Diagnostics
Resubmission - Manuscript ID: diagnostics-2656788
Dear Editor,
I would like to resubmit our manuscript titled ‘Expression of preferentially expressed antigen in melanoma, a cancer-testis antigen, in carcinoma in situ of the urinary tract’ for publication in Diagnostics.
The reviewers’ comments were highly insightful and enabled us to greatly improve the quality of our manuscript. Please find below the point-by-point responses to all the comments provided by the Reviewers. The revisions in the manuscript can be seen in red color font.
I hope that with these revisions you will find our manuscript closer to publication in Diagnostics. I look forward to hearing from you at your earliest convenience.
Sincerely,
Mitsuaki Ishida, MD, PhD.
Department of Pathology, Osaka Medical and Pharmaceutical University
2-7, Daigaku-machi, Takatsuki City, Osaka, 569-8686, Japan.
Tel: +81-72-683-1221
E-mail: mitsuaki.ishida@ompu.ac.jp
Response to the comments of Reviewer #2
Thank you very much for reviewing our manuscript. We appreciate your constructive comments. We have made the following revisions in response to the issues you raised.
- This is rather a small cohort, the authors should at least highlight this limitation in the discussion.
Response: Thank you for pointing this out. As you suggested, we added the limitation regarding the cohort size in the Discussion section.
“First, this was a single-center retrospective analysis with relatively small number of CIS patients, potentially leading to bias in the statistical power. Therefore, additional multi-center studies with larger cohorts are needed.” (line 299-302)
- Figure 1 is not intuitive, I suggest authors to manually highlight those cells with distinct morphology so that readers with limited pathology background may understand.
Response: Thank you for your suggestion. We changed the Figure 1. It now shows the typical histopathological features of CIS with a higher magnification.
- What is the criterion for determining whether a patient is positive or negative using IHC?
Response: The criteria of the immunohistochemical stainings of PRAME and p53 were the same as those adopted in the previous reports, and are stated in the section of 2.3 Immunohistochemical analyses.
“Positive immunoreactivity for PRAME, manifesting in either nuclear and/or cytoplasmic positivity, was defined as any expression in carcinoma cells of CIS, employing the same methodology as that outlined in the previous report [31]. Similarly, p53 immunoreactivity was assessed in accordance with the previous report [35], as follows: aberrant patterns included either diffuse positive (with the majority of carcinoma cells exhibiting homogeneous moderate to strong positivity) or negative/null patterns, while wild-type patterns featured patchy positive cells.” (line 140-146)

Round 2
Reviewer 1 Report
Comments and Suggestions for Authors
I want the authors to provide p value regarding the correlation between PRAME expression and CIS
Author Response
Response to the comments of Reviewer #1
Thank you very much for reviewing our manuscript. We appreciate your constructive comment.
I want the authors to provide p value regarding the correlation between PRAME expression and CIS
Response: Thank you for your comment. In the present study, PRAME expression was noted in 23 of 53 patients of CIS (43.3%), but none of the non-neoplastic urothelial epithelium around the CIS lesions in all cases. As you suggested, we performed statistical analysis of PRAME expression between CIS and non-neoplastic urothelial epithelium, and added the comment. (3.2. Histopathological and Immunohistochemical Features)
“PRAME expression was noted in 23 of 53 patients (43.4%), but non-neoplastic urothelial epithelia around the CIS lesions showed no positive immunoreactivity for PRAME in all cases (p<0.001). Positive immunoreactivity for PRAME in CIS was noted in the nuclei and/or cytoplasm of the neoplastic cells of CIS, mostly only in the nuclei of the neoplastic cells.” (line 168-172)
Round 3
Reviewer 1 Report
Comments and Suggestions for Authors
Statistical analysis is now complete